# The Unwanted Heritage of Prefabricated Wartime Air Raid Shelters—Underground Space Regeneration Feasibility for Urban Agriculture to Enhance Neighbourhood Community Engagement

Paweł Matacz [1] and Leszek Świątek [2,*]

1    Paweł Matacz Architecture, Somosierry 6, 71-179 Szczecin, Poland; paw.matacz@gmail.com
2    Faculty of Architecture, West Pomeranian University of Technology in Szczecin, Żołnierska 50, 71-210 Szczecin, Poland
*    Correspondence: Leszek.Swiatek@zut.edu.pl

**Abstract:** The article deals with the problematic heritage associated with the system of Nazi German underground air raid shelters currently located within the Polish state, in the Baltic port city of Szczecin. The unwanted heritage has been inventoried, archival materials collected, and comparative analyses made of ways in which the underground space can be revitalised. An attempt was made to develop a typology of existing shelters and their locations. In order to overcome the negative associations with the warlike military space, positive system solutions were sought for the productive use of existing concrete structures located underground in central, easily accessible areas of the city districts. A process of upcycling the space was used to make ecologically efficient use of the material resources contained in the shelters. In order to activate the local community, a modular, hydroponic plant-growing system, adapted to the prefabricated spaces of the historical air raid shelters, was proposed. In this way, the central location of the underground structures within the boundaries of residential neighbourhoods was exploited. Such action strengthens the food sovereignty of the inhabitants, initiates bottom-up activity within the boundaries of the neighbourhood unit, and builds social ties in the spirit of a regenerative economy and positive sustainability.

**Keywords:** underground space recycling; heritage management; urban farming; city aquaponics; regenerative culture

## 1. Introduction

The unwanted and forgotten heritage of World War II—Nazi underground air raid shelters—has become a planning challenge for the dynamically developing city of Szczecin, Poland's important harbour, academic centre, and emerging new technology industry. The large number of military shelters located in the central parts of Szczecin's districts provided a sense of security for the then-German inhabitants of the city during carpet bombing by the Allied Forces [1]. After the war, due to border changes in Europe, Szczecin became a Polish city, receiving many migrants from Lvov or Vilnius, which were occupied by the Soviet Union [2]. Taken over after the war, the destroyed city was an enemy city. The new inhabitants had negative associations with the existing Nazi military infrastructure. As Paul Virilio describes in his book "Bunker Archeology": " I would ask if people still had the opportunity to study other cultures, including the culture of adversaries—if there were any Jewish Egyptologists. The answer was invariably, "Yes, but it is a question of time... time must pass before we are able to consider anew these military monuments" [3] (p. 14).

Szczecin, similarly to many other European cities, has a rich history which cannot be linked to just one nation. Over the centuries, many different peoples and cultures have been connected with the city. In the case of Szczecin, we can mention the Slavs, Poles, Germans, Prussians, Swedes, Danes, and Jews. They have all contributed to the way the city functions

and looks today, so it is difficult to attribute the entire history of the city to a single nation [4]. It is reasonable to question the view of the city's history as a city with only Polish roots, but rather to treat it as layered, heterogeneous, multicultural structures that were built on top of one another. Each layer is essential to the existence of the other, making them equally important. For most of Szczecin's history, the city grew and developed. Despite changing nationalities, the number of inhabitants increased. Szczecin was part of German territory until the end of World War II. It was a city with a large synthetic fuel factory and a port connected to the Baltic Sea. It is also an important crossing point of the Oder River. This situation lasted until the outbreak of World War II, when the number of inhabitants rose sharply due to the strategic nature of the city. In 1939 the population reached 380,000 [5]. In order to protect such a large number of people, the construction of small air raid shelters and the adaptation of existing underground spaces for shelter purposes began on a large scale [6,7]. The underground system, consisting of about 370 shelters and 17,000 cellars, could accommodate almost half of the city's population [7]. Due to its strategic importance, the city had to be captured before the further offensive on Berlin. Using air superiority and the poor condition of the German Luftwaffe, the Allies aggressively bombed the cities before the Red Army's ground offensive. In 1943 and 1944, the Allied forces carried out a series of 26 bombing raids, lasting a total of over 283 h [1,4,8,9]. The alarm system was activated more than 300 times. Night air raids dropped bombs, blindly annihilating the old city and destroying other large parts of the city. For the inhabitants of the city, the shelter system played a crucial role in their daily lives. Despite the efforts made to protect the inhabitants of Szczecin, some 46,000 people lost their lives during the bombings between 1943 and 1944 [10–13]. The system potentially saved hundreds of thousands of lives, and in the case of some districts witnessed the destruction of 90% of the city (Figure 1) [14,15].

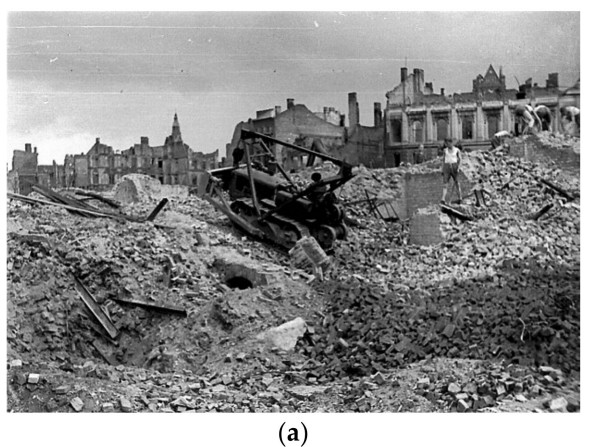

(**a**)

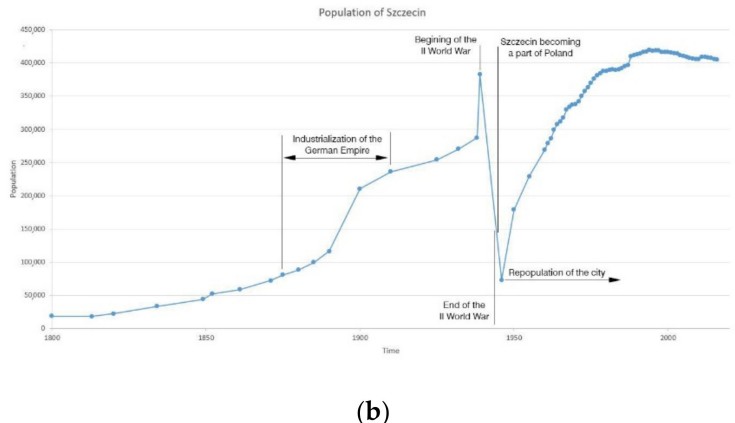

(**b**)

**Figure 1.** The civilian air defence played a key role in the history of the city, and as a cultural heritage resource it should be revitalised in a way that makes its historical character recognisable and ensures the contemporary restoration of the social value of the spaces found [16]. (**a**) Cityscape of the outer bailey after the carpet bombing of the allied forces; (**b**) the period of the collapse of the population of Szczecin during the air raids at the end of the Second World War [5,8].

Today, the architectural heritage of former nations is mostly divided in the minds of the city's inhabitants. One group consists of assimilated buildings that have been accepted as part of the city and are kept in good condition, such as Wały Chrobrego (Chrobry Embankment), Zamek Książąt Pomorskich (Pomeranian Dukes' Castle), St. Peter and Paul's Cathedral, Jasne Błonia, and Park Kasprowicza (Kasprowicza Park) [17,18]. The construction of all of these can be traced back to the Germans, but for many their origins are largely unknown or irrelevant, as in their minds these buildings are already part of contemporary Szczecin. The second group is made up of the most devastated and neglected buildings. This group includes buildings such as the Quistropp Tower, the Bismarck Tower, and the system of civilian shelters [19]. In relation to these objects there is usually a clear

reference to their Germanic origin. In the case of the shelter system, the most common association is with the military and times of war. It is not uncommon to hear the terms "bunkers", "Old German bunkers", "Nazi bunkers", or "Hitler's bunkers" [20,21].

On the general timeline, underground civil defence infrastructure has been placed more recently and has a short life span compared to other historical underground spaces, such as underground cities and burial sites or underground water infrastructure. The willing use of these military spaces by civilians has also not been dictated by any factors other than the objectives of ensuring their own and their loved ones' safety and survival. It has a much different relationship to its users than any other space, because this relationship is based solely on fear.

One of the challenges awaiting future revitalisation processes of air raid shelters is the specific emotional connection people have with underground military spaces [3]. Although in many cities the underground has lost its negative historical connotations, largely due to the development of underground infrastructure, in Szczecin and other Polish cities the publicly accessible underground is incidental, hence its perception is negative [22,23]. Combined with their narrow and low-rise buildings with many labyrinthine nooks and crannies, as well as a specific microclimate, they can evoke anxiety or fear associated with the Second World War period (Figure 2).

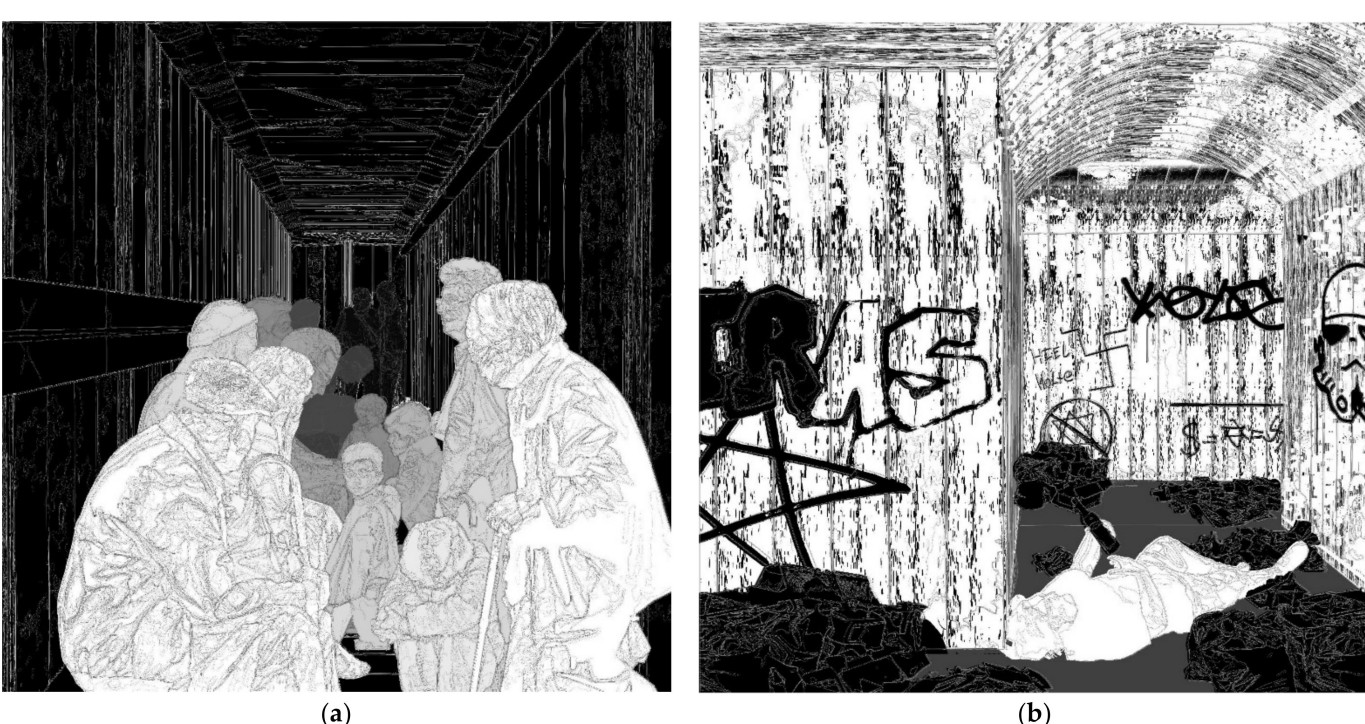

(**a**)        (**b**)

**Figure 2.** Visualisations. (**a**) Inside of the shelter during air raid. (**b**) Representation of the current state of degradation of shelters.

## 2. Materials and Methods

For the purposes of the presented project, historical and archival research was carried out, interviews and inventories of selected sites were conducted, the urban space mapping method was applied, and social analyses as well as comparative studies were undertaken. The general structure for the research, consisting of 9 steps, was established as shown in Table 1.

**Table 1.** Selected research methods and objectives.

|   | Research Methods | Objectives |
|---|---|---|
| 1 | Interviews with inhabitants | To understand the importance of and collect opinions about the shelter system and establish its position on the mental map of the city of its inhabitants |
| 2 | Literature research | To understand the historical background and to collect information about the scale and size of the system |
| 3 | Publication research | To establish the value and potential of the system |
| 4 | Archival research | To determine the spatial qualities of the underground spaces and understand their technical aspects |
| 5 | Site visits | To confirm the previously collected data about the placement of the facilities and their construction. To collect information about their current state |
| 6 | Comparative research | To collect and compare different approaches to revitalisation of similar spaces |
| 7 | Interviews with activists, engineers and representatives of municipalities. | To establish a socially, financially, and legally feasible approach to revitalisation of the shelter system in Szczecin |
| 8 | Concept generation | To confirm if it is possible to propose a design for the revitalisation of the entire system |
| 9 | Preliminary design for the pilot location [1] | To test if the previously planned concept can be applied in one of the locations |

[1] For the creation of coherent and readable graphics, multiple software programs were used. The technical drawings were made using ArchiCAD and the graphs were generated using Microsoft Excel. Visualisations were created and edited in Adobe Software based on a 3D model built in SketchUp.

Most information on the history of the city and the underground shelter system was found in available publications, books, and the National Museum of History of Szczecin. Stettin's civil defence structures built during World War II are not recognised as monuments, and no information on them can be found at the Provincial Office for the Protection of Monuments in Szczecin. There is also no up-to-date documentation of these spaces, and most of the information that can be found comes from incomplete German archives or declassified documents relating to the organisation of Anti-Aircraft Field Defence, which was the body responsible for the organisation of civil defence between 1951 and 1964 [19]. A complete and coherent study of the underground civil defence system in Szczecin is difficult to achieve due to the confidential and secret status of some military and administrative documents in the past and the institutional dispersion of competences today.

In order to define the situation in the context of the entire city, a map with the location of 27 air raid shelters was created. The map was based on archival materials of the Presidium of the National Council, historical publications, and field visits. The map was analysed in terms of the distribution of facilities in the city, access isochrones, and the relationship of shelters to each other. Based on the results, several locations were selected, and further in-depth analysis was carried out on a district scale to identify possible local factors.

The next step was to visit the selected sites to determine if the archival data found were correct as to their location, to check the condition of these spaces, current informal use, state of preservation, and degradation. The social studies were aimed at identifying possible development potentials. The research started with an analysis of the spatial development plans of the city of Szczecin and the surrounding regions to see if the local authorities perceived any historical, social, or tourist potential of the spaces in question. Research carried out by the University of Gdansk on social groups potentially interested in visiting the underground spaces was then taken into account and analysed together with data collected from tours in the district [24]. The final stage of comparative research consisted of collecting examples of successful revitalisations of military shelters and comparing them with each other in terms of the quality of preserved space, historical reserve, community building potential, current contribution to the local community, inclusiveness, and sustainability.

To establish the legal framework for the project, several experts were briefly consulted: architects, a representative of the Provincial Conservator of Monuments, a health and safety expert, a representative of the Civil Defence, and a local journalist reporting on underground Szczecin. The aim of the study was not to discuss all the regulations that could affect the design process, but rather to highlight possible obstacles.

The collected information was then analysed and a framework as well as key guidelines were established. Based on those conclusions, a concept for the revitalisation was created and presented to the public in order to collect feedback, which was then applied to the project. To test the applicability of the improved concept, a project of the adaptation for one chosen location was created and tested to see if it meets the key legal requirements for such an adaptation. It was then publicly displayed and presented to representatives of the architectural community of Szczecin in order to collect other feedback, this time focusing on legal and technical aspects of the design which were then applied to the final project. The finished product of the design process was then analysed and compared to the previously found examples of similar revitalisations, and then discussed.

### 3. Results

#### 3.1. Findings of the Inventories of Available Underground Air Raid Shelters

Currently, most of the entrance pavilions are partially destroyed: the majority of them are lacking the original metal doors, and the entrances have instead been bricked up by the municipality to prevent unsupervised entrances. In some cases steel bars were mounted in place of the doors if the shelter was recognised as a habitat of bats, which are under strict protection in Poland. The destruction of some of the pavilions could be linked to the bombings itself, others to long term exposure of the raw structure to the temperature seasonal climate of Central Europe. Some bear traces of forced entries and break-in attempts. The shelters that have some possibility of entry are usually filled with trash and heavily vandalised; it is not uncommon that they serve as winter shelters for the homeless, or meeting spots for representatives of some rare youth subcultures. The conditions in the tunnels itself are extremely diverse; there are tunnels that are still in good condition, but others suffer from long exposure to moisture or in some cases are flooded because of the lack of maintenance of the drainage system. Their structural safety also represents a wide range. Many tunnels are collapsed or partially filled with sand, which is the case in the shelters that were never completed, and end with unsealed tunnels of wooden construction (Figure 3).

The rapid expansion of the system during the war needed to provide shelter to an increasing number of civilians, and therefore required the implementation of solutions that were easy to build, cheap, and repeatable. For this reason, designs of "typical shelters" were introduced. These highly adaptable designs, based on simple technologies, allowed for the construction of such structures on almost any plot of land, regardless of its size and ground conditions. It was not uncommon to begin construction work without first determining the final shape or capacity of the facility. If the workers encountered problems with excavating the earth, they would change the direction of the tunnel; if it was not possible to dig any further, the tunnel simply ended at that point [10,25,26]. This flexibility was achieved by using a modular design approach that focuses on the functionality of the facility regardless of its shape or size. This systematic approach allows existing structures to be classified without analysis of each one, and similarly flexible redevelopment solutions based on the principles of modularity, adaptability, and flexibility to be prepared before costly and time-consuming spatial and structural building surveys are carried out. Analysing a selected group of over thirty underground shelters, it can be concluded that despite the great variety of forms they all share a common tunnel scheme. The most common solution was a long meandering corridor with two exits equipped with airtight doors. They were usually placed shallowly underground, either entirely below ground level using excavated material from the construction site, or partly below ground level using excavated material to make an additional protective and camouflage layer on top of the shelter, forming a

small mound [10,25,26]. Both structures are barely visible from the air even with modern technology, and their location is only betrayed by a slight shadow from the embankment and a slight change in colour of the vegetation on top of them.

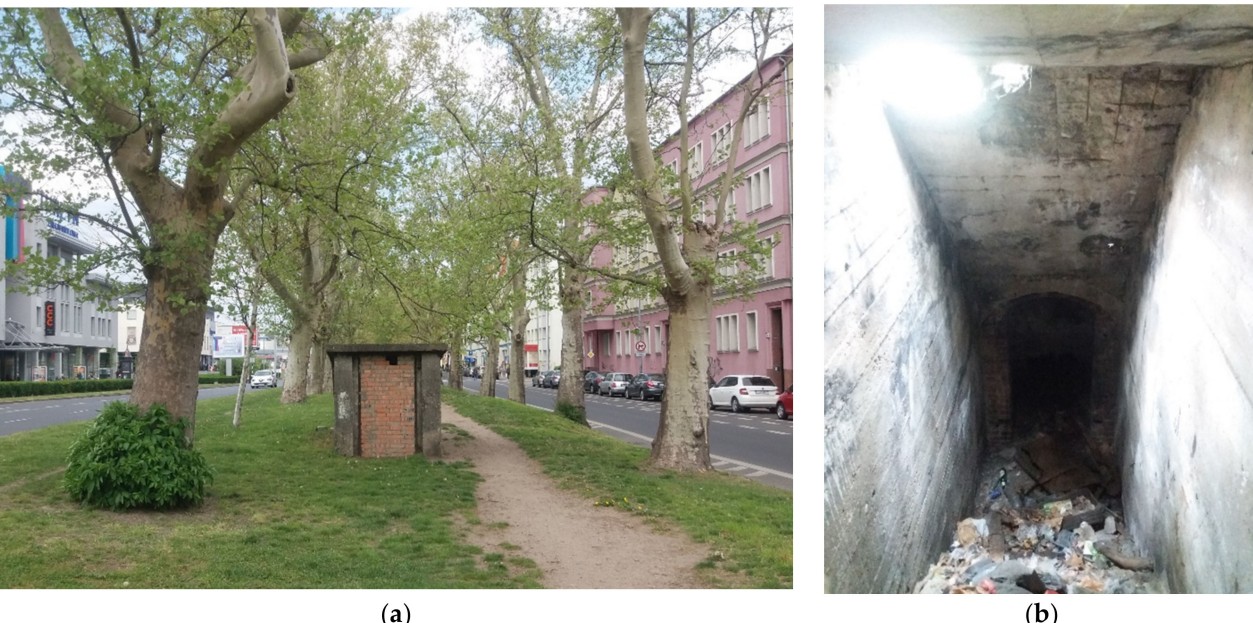

(**a**)  (**b**)

**Figure 3.** Present existence of air raid shelters. (**a**) Location in the city fabric. (**b**) Current state of the entrance zone.

A characteristic feature of the World War II period is the dispersion of shelter locations on the scale of the city and its suburbs. The chosen model for the construction of shelters is a consequence of both the spatial development of Szczecin and the strategy of total war, based on a massive scale of air raids and bombardments. The decentralised 20th century air raid shelter system contrasts with the central location of bunkers and fortifications built in the city in the 17th and 18th centuries. They differ not only in location but also in form, speed, and technology of construction, as well as in the type of users for whom they were intended to provide safe shelter (Figure 4).

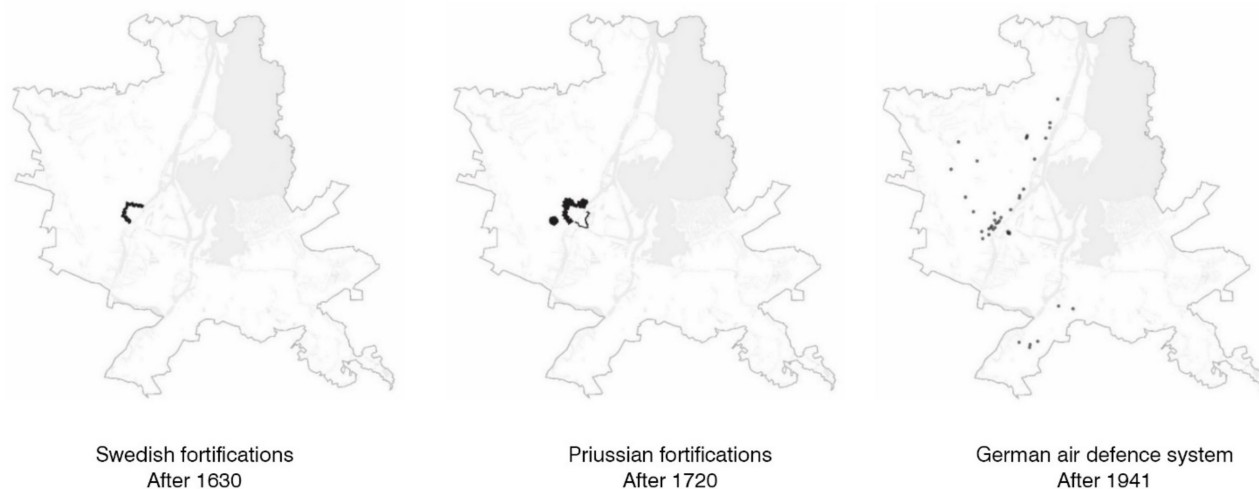

Swedish fortifications
After 1630

Priussian fortifications
After 1720

German air defence system
After 1941

**Figure 4.** From a compact, centralised pattern to a dispersed one—the changing system of fortifications and shelters in the city's history in the context of Szczecin's current administrative borders.

*3.2. Types of Shelters Analysed and Their Characteristics and Classification*

All facilities based on corridors can be divided into two groups:

- Organic, in which the tunnels curve slightly and the overall layout is less ordered. Shelters with a complex organic plan were built with cradle vaults, which were either poured in situ with concrete or laid with bricks. The material depended mainly on the time of construction: brick tunnels are mostly remnants of the pre-war fortifications of the city, whereas weather-cast structures can be dated to the early years of the war (Figure 5).
- Geometric, where the tunnels are straight and most of the bends are at right angles. These shelters were built from prefabricated reinforced concrete elements that could be joined at certain angles (Figure 5).

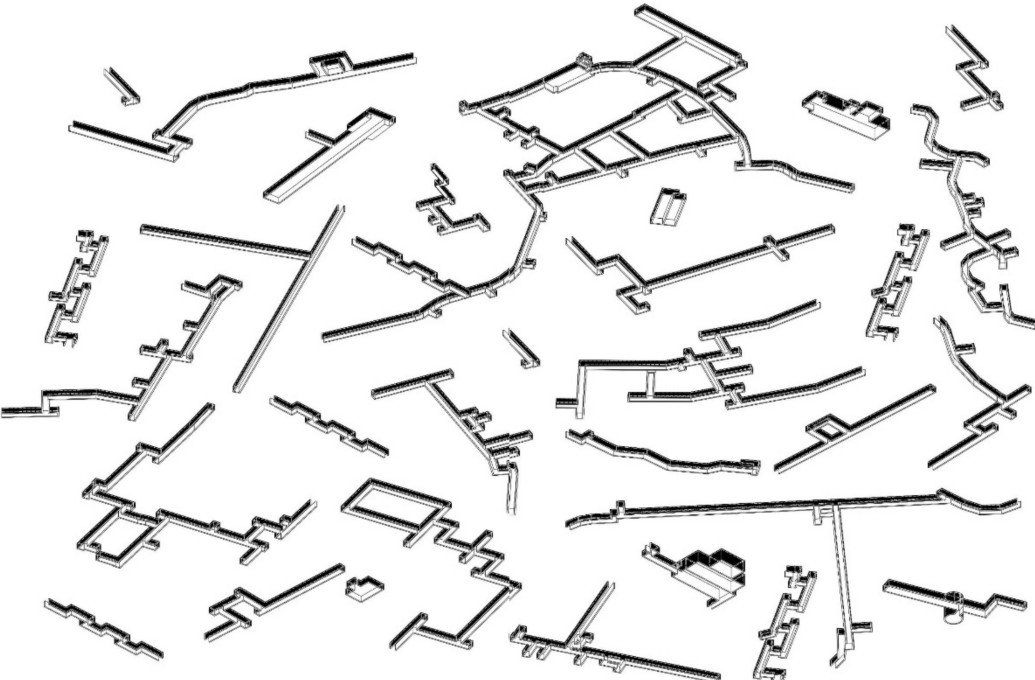

**Figure 5.** Catalogue of underground spaces [10,25,26].

3.2.1. Heinemann Prefabrication System of Air Raid Shelters

A system design for a prefabricated shallow underground air raid shelter has been preserved in the State Archive in Szczecin. The documentation was produced by the Berlin firm Heinemann & Co. in 1942 and consists of plans, cross-sections, construction details of each prefabricated element, and material cost estimates. The whole system mainly consists of three basic prefabricated elements:

A floor slab 8 cm thick, 142 cm long, and 25 cm wide, weighing 70 kg. The slab was reinforced in its upper part with two 7-mm-thick iron bars. Additional reinforced trapezoidal projections 4 cm high have been added to both ends of the slab to hold the side walls in place.

The wall plate is 8 cm thick, 195 cm high, and 25 cm wide, weighing 100 kg. The slab was reinforced with three 8 mm iron bars on the inside, one 6 mm iron bar on the outside, and a 15 mm stud hole in its top.

A ceiling slab 10 cm thick, 142 cm long, and 25 cm wide, weighing 90 kg. The slab was reinforced with two 8 mm iron bars on the inside and one 6 mm bar on the outside. As with the floor slab, it had additional reinforced trapezoidal projections 4 cm high on both sides to resist soil pushing against the walls.

The concrete structure was found in a 1.48 m deep trench, and the excavated material was placed on the shelter in the form of an embankment 0.90–1.15 m high and 1.52 m

wide, with sides sloping at 45 degrees, with an additional 25–50 cm layer of earth on the shelter slab.

### 3.2.2. Layout and Realised Shelter Configurations

The documentation includes four possible configurations of shelters based on the desired number of users, demonstrating the high adaptability potential of the system:

- One corridor with two entrances for 50–60 people;
- Two corridors with three entrances for 100–120 people (Figure 4);
- Three corridors with three entrances for 160–180 people;
- Five corridors with three entrances for 200–240 people.

The corridors connect into 60-60-60 triangular nodes; the third side of each node was used either for a short tunnel leading to the entrance or for a small niche that could be equipped with sanitary facilities such as sand-flush toilets. Each corridor was only 12 m long, which prevented the shock wave from spreading and reduced casualties in the event of a direct bomb hit, as these shelters were designed mainly to reduce civilian casualties during air attacks rather than to fully protect them.

### 3.2.3. An Alternative Systems of Air Raid Shelters

However, this is not the only system used. Another variation of this system, which required the use of cast-in-place concrete, was also widely used. It combined the flexibility and speed of building precast systems with the rigidity offered by permanent concrete joints. This system consisted of three precast elements: a floor slab, a wall slab, and a ceiling slab (Figure 6). However, each of these elements was fitted with curved iron bars extending beyond the concrete at their sides. This reinforcement interlocked to form an iron crest at each of the four internal corners of the corridor section. Once all four elements were in place, a single-panel formwork had to be cast to cast a permanent connection to reinforce the structure. Both systems made it possible to conveniently build the tunnels metre by metre; this is particularly evident in tunnels excavated using traditional mining techniques. Many concrete passages end abruptly and are followed by an earth tunnel several metres long, protected by wooden stakes and planks. At the end, usually where harder rock has been encountered, pick marks can be seen.

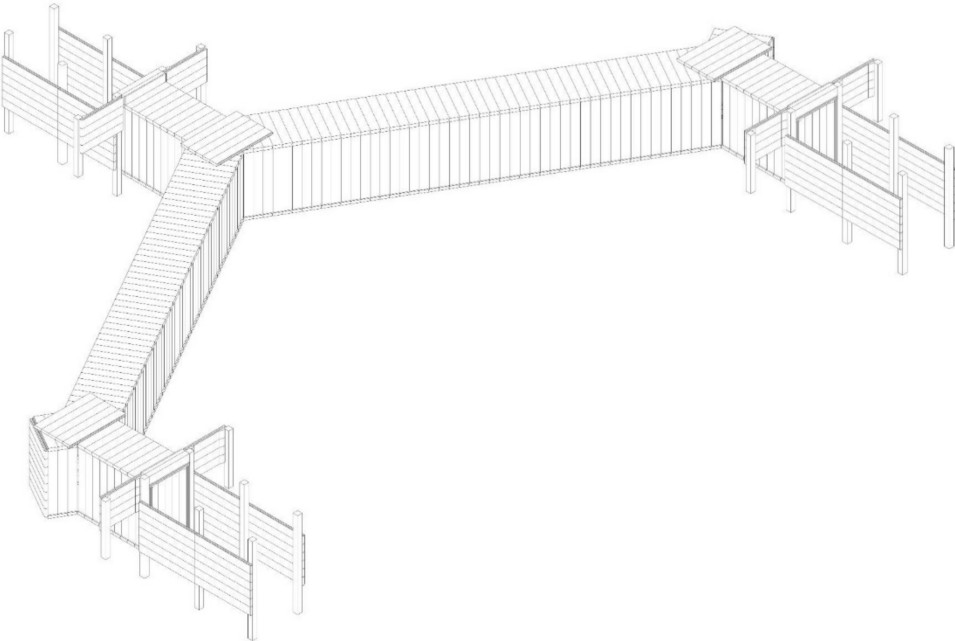

**Figure 6.** Two corridors with three entrances for 100–120 people.

### 3.3. The Entrance Zone as a Connector between Different Realities

The entrance part was a symbolic gate between the world of the blue sky and the depths of the underworld. Many different entrances can be found in different shelters. Their design, number, and location depend mainly on the year and type of shelter. In the case of small shelters built during World War II, however, certain trends can be discerned. Similarly to the shelters themselves, the tunnels were designed and built in a systematic way. The technical documentation found in the archives did not contain details on the construction of entrances. In contrast to the tunnels, the entrances are widely accessible from the public spaces of the city and observation is possible. Based on the data from numerous field visits, a categorisation of entrances can be proposed.

Above-ground entrances are the most common. They are most often found in facilities located in parks or in green belt that divide street lanes in the city centre. They consist of prefabricated concrete covered pavilion stairs leading down to the main tunnel doors. The stairs are slightly wider than the tunnel itself and were originally protected by heavy steel doors. Today, many of these doors are gone and the entrance is bricked up instead. There are two types of these entrances: a small, triangular one in side view, consisting of a sloping panel over the stairs and door, and a larger, trapezoidal one in side view, consisting of an addition of a horizontal panel over the upper platform of the stairs. It is not clear under what conditions the decision was made to use one type or the other; perhaps it was related to the available space in front of the tunnel when it was built. Today, both types of tunnels usually have a large amount of open space in front of them, sometimes interrupted by existing pavements or paths, or roads in a few cases.

A less common type of entrance is characterised by the absence of a concrete pavilion. Instead, the stairs leading to the main door located underground are protected by a steel door placed flat on the ground surface. The flat placement of the door is made possible by the deeper layout of the corridors and the absence of an earth berm on top of the shelter.

The third category consists of vertical entrances, usually found in shelters located within natural escarpments such as hills or artificially made embankments. The entrance consists of a steel door surrounded by concrete reinforcement placed in the surface of the slope.

For the purposes of future revitalisation, we can distinguish 24 different currently existing location types. This division takes into account the type of entrance, the state of preservation, the location in relation to the ground level, and the existing space in front of them.

### 3.4. Case Study Comparison of Adaptation Processes and Transformation for Different Types of Military Shelters and Target Groups

The adaptations of unused underground military spaces are an example of upcycling, a form of recycling that focuses on reusing and repurposing entire objects, as opposed to downcycling, which focuses on reusing materials or elements of an object [27,28]. It is an environmentally friendly way of dealing with abandoned spaces that have already lost their potential to fulfil their original function.

In many cases, other levels of sustainability can also be achieved, such as the preservation of cultural heritage through self-financing, which leads regeneration to further levels of socio-economic or political sustainability by promoting historical awareness [29].

Among all adaptations of former military shelter facilities, we can distinguish some trends.

#### 3.4.1. Transformation Capacity for the Residential and Office Sectors

The first group consists of residential and office adaptations. One such example is the German city of Siegen, where architects from Modulbüro have proven that it is possible to revitalise the thick concrete walls of an air raid shelter into spacious luxury flats [30]. This is not an isolated example: another German company, Bunkerwohnen, focuses its efforts exclusively on adapting shelters for residential purposes, with six successful attempts to its credit [31,32]. In the US, the market for flats located in shelters is also booming,

with developers adapting Cold-War-era buildings, fuelling their marketing with promises of security capable of withstanding a nuclear strike [33]. All the above examples are adaptations of large facilities. However, most of the civilian anti-aircraft system in Stettin is decentralised and consists of a large number of small sites, so strategies that have worked well for large sites may not be appropriate here. In the Dutch town of Vuren, architects from the Belgian office B-ILD made a successful attempt to transform a small bunker, part of the New Dutch Waterline, into a holiday home [34]. The cramped spaces of the shelter were compensated for by connecting it to the green surroundings, where a large terrace was created. The contrast created combines the best of both spaces and provides a unique spatial experience. Another notable example in this category is the headquarters of the Swedish Bahnhof AB in Stockholm, designed by Albert France-Lanord architects [35]. This medium-sized office is located in a nuclear fallout shelter 30 m underground, and is a great example of reusing existing spaces rather than further densifying or spreading out the city.

### 3.4.2. Examples of Adaptations for Museums and Exhibitions

The second group consists of adaptations for museum purposes and quasi- and pseudo-museums. Military objects, due to their historical value, are perfectly suited to house museums. Authentic spaces are a value in themselves, providing a special experience for visitors and an excellent background for exhibitions focusing on their periods. One of the most famous examples is located in the Danish town of Blåvand. BIG designers have added almost three thousand square metres of underground exhibition space to a bunker that was part of the Atlantic Wall [36]. The facility houses four independent museums, which together form a historical educational complex. However, not all adaptations are carried out with care for historical authenticity. In 2006, the Russian Federal Agency for State Property Management privatised Bunker 42 in Moscow [37]. According to the official website, the bunker has been converted into a museum, but in reality, it offers an experience in the form of a theme park focused on the pop culture image of the USSR. It consists of conference and event rooms, a small cinema, a bar, a wedding hall, and is soon to be expanded with a spa area. These two contrasting examples show the diversity of approaches to active historical education. The Dutch bureau Rietveld Architecture-Art-Affordances took a different approach when renovating one of the New Dutch Waterline bunkers [38]. By radically cutting the small structure in half, they exposed the imposing and invisible mass of its construction. The experimental intervention defies current conservation practices, prompting a debate about the current methodology. The exposed structure has been declared a national monument and open to all, which opposes active education and focuses attention on the structure itself.

### 3.4.3. Adaptations for Commercial and Social Purposes

The third group is commercial and social adaptations. The two largest military shelters in Hamburg have been adapted for social purposes. The smaller one, built in 1943 and partially destroyed in 1947 after the demilitarisation of Germany, was converted into a zero-emission power plant, providing energy for 50,000 inhabitants. The adaptation has also played a role in promoting sustainable development strategies, winning numerous architectural awards and participating in the German exhibition at the Venice Biennale in 2008 [31]. A larger shelter built in 1941 has been adapted into a community centre, housing galleries, studios, a music school, and numerous shops [39]. At night it serves as a club. In the near future the bunker is to be extended with a large open-terrace garden. Another noteworthy example is the KEBAP project, a grassroots, community-driven attempt to transform another of Hamburg's shelters into a climate-neutral, wood-powered power station with the added function of a community centre [40]. A more commercial approach was taken by London-based Growing Underground [41]. This company used one of the tunnels of an anti-aircraft system in the borough of Clapham for plant production. Using new technologies, the underground spaces have become a powerful site for plant production. Separated from the environment, they provide control over growing conditions

not achievable on the surface, and by using low-energy technology they are more energy efficient than surface production because there is less energy loss. The company supplies local markets with locally sourced vegetables and salads, significantly shortening the supply chain and making food more environmentally friendly and fresher. The project has been a marketing success and has gained a lot of support from the local community.

### 3.5. Building Empathy between the Creative Potential of Underground Spaces and the Target Group of Users

Each of these transformation or regeneration examples targets a specific user group. The examples of housing projects, seemingly intended for everyone, filter the actual users according to their income. High adaptation costs combined with historical values and unique aspects of the space drive the price above the market average. A similar filtering of users occurs for exhibition spaces. Widely considered to be inclusive, especially when co-financed by state budgets, which reduces the income factor as the spaces are freely available, they tend to contain a threshold of commitment or a minimum of education, excluding a section of the population with low income and low education.

However, most revitalised cases of underground military spaces in Poland focus on their military aspect, which naturally narrows the possible spectrum of their users.

According to extensive research carried out by the Institute of Geography of the University of Gdansk, underground facilities in Poland have a narrow target group of users. Not only military spaces, but all areas of underground heritage, including sites such as mines and natural rock formations were surveyed. Within this broad spectrum of sites, men account for almost 80% of all visitors. If one takes into account the deep-rooted culturally masculine nature of militaria, an even higher percentage of men in underground military spaces is very possible. The largest proportion of visitors has a secondary education, and those with a university degree and primary education are a distinct minority. Usually, the only way for minors to access the underground is under adult supervision, usually in the form of educational school trips [24]. These, however, generally must be organised to curriculum-related facilities. Unauthorised access to most underground military spaces is also not permitted for adults. Each tour usually must be planned and takes place in small groups, which must be accompanied by a guide. Such a restrictive system can discourage spontaneous visits and, together with the fact that most visitors are of working age, contributes to overcrowding and bottlenecks at weekends in addition to under-use on weekdays.

In the past, when the shelter system was used according to its original function, apart from the enormous racial, ethnic, or religious discrimination associated with Nazi National Socialist ideology, the spectrum of its users was much wider than it is today. Their use was of course forced, as people had little choice in the face of the threat of a bomb hit, but nevertheless the only factor determining their users was the distance they lived from the shelter. Of course, any contemporary revitalisation of underground air raid shelters is associated with a kind of narrative and a game of imaginary space. As Virilio noticed: "This game created an implicit empathy between the inanimate object and visitor, but it was the empathy of mortal danger to the point that for many it was unbelievably fearsome. The meaning was less now that of a rendezvous, and more of combat: If the war were still here, this would kill me, so this architectural object is repulsive." [3] (p. 14).

### 3.6. Proposal Description of Prefabricated Wartime Air Shelter Regeneration

The first and most critical step in building a revitalisation plan is to select a function that could exist given all the constraints analysed and contribute to achieving the objectives. For this purpose a range of selected functions, based mostly on analysed case studies, were taken into consideration. Residential and office functions were discarded based on the scattered character of Szczecin's system and the size of the facilities. For a similar reason functions linked to art were discarded. Provided space could only serve as a limited gallery. A passive function, combined with low demand and limited range would significantly reduce chances for successful revitalisation. We have found commercial

and social functions to be the most promising, preferably combined. Amongst retail, information centres, education facilities, farming, oxygen distribution facilities, temporary homeless shelters, and energy storages, urban hydroponic farming was the most inclusive and presented a range of social benefits. It is an easily scalable and well-researched function that helps in the creation of intergenerational social connections, encourages involvement and local appropriation, stimulates sense of belonging, and provides physical and easily countable rewards in short cycles.

### 3.6.1. Phenomena of Urban Agriculture Issues

In this case, urban agriculture fits the best. In addition to the most basic benefit of local food production, it also serves an educational purpose, strengthens the local community and overall civic engagement, and combines with the benefits of reusing historic spaces, such as an increased sense of belonging and historical awareness [42]. However, traditional agriculture requires large tracts of land and good exposure to daylight, something that air raid shelters inherently lack.

Both problems can be solved with the implementation of hydroponic systems, which allow plants to grow sustainably without the use of soil. First published in 1627 by Francis Bacon, the system has achieved huge popularity in the last decade and has entered an exponential growth phase. The hydroponic food production market was estimated at 1.33 billion in 2018 with an estimated growth of 22.5% (2019–2025), compared to the 5% growth (2020–2027) of the global food market. Widespread adoption of the technology has impacted the private sector with the introduction of affordable pre-made systems for growing vegetation at home [43,44].

The basic systems consist of four subsystems: ventilation—providing optimum temperature and air composition; lighting system—using energy-efficient LED lighting capable of adjusting the wavelength according to the growth phase of the plant; nutrient system—consisting of a water tank and separate nutrient stores; and a central control unit—controls the entire process with the aid of numerous sensors. Regardless of advances in farm mechanisation, the human factor is still crucial in this process.

Food production using completely artificial lighting, regardless of the technology used, consumes large amounts of electricity. This fact generates debate and is used in favour of traditional or greenhouse farming. However, in order to compare these methods, more factors need to be taken into account. According to a study by the Swedish Environmental Institute, which used the LCA method to assess the environmental impact of greenhouse and hydroponic farming, in addition to higher light production costs, hydroponic farms use less energy to control temperature and humidity levels [45]. They also use significantly less water. Given the growing global demand for water and the fact that the food industry contributes to 60% of global consumption of this limited resource, this argument may be much more important, as we can easily produce electricity. Hydroponic crops also offer 30% higher yields and have the advantage of being produced within cities, shortening the supply chain, reducing the use of non-renewable resources, and reducing greenhouse gas emissions [45].

### 3.6.2. Harmonisation of New Prefabricated Hydroponic Equipment Modules with the Existing Space of Wartime Prefabricated Shelters

In order to design a module that would fit all the different systems used to construct underground air raid shelters, a comparative analysis of their cross-sections was carried out, which resulted in the identification of the common space available in the cross-sections of all the tunnels. A modular system that fits within the limits of the common space will work properly in each of the tunnel shelters. The available space is 195 cm high and 120 cm wide. In each corner of the section a triangular space was subtracted to fit the connectors: in the lower corners—16 cm wide and 12 high; in the upper ones—28 wide and 22 high. For the purpose of vertical hydroponic cultivation, this space is sufficient.

The cardinal principle of the project is to divide the corridor into spatially identical fragments, which will be able to join to form longer structures, just as in the original tunnel

construction scheme. Each fragment will be 2 m long, which corresponds to the length of eight prefabricated structural elements and 1/6 of the length of a standard 12 m corridor, and will be the basis for the design of a standardised module to accommodate all the shelters (Figure 7). This module will then be divided into two parts: a 30-cm-deep production space and a 75-cm-deep routing space. This division was based on the minimum space required for the storage of pre-assembled hydroponic systems available on the private market and the minimum space specified in Polish health and safety regulations for one-way access to the equipment in facilities intended for production.

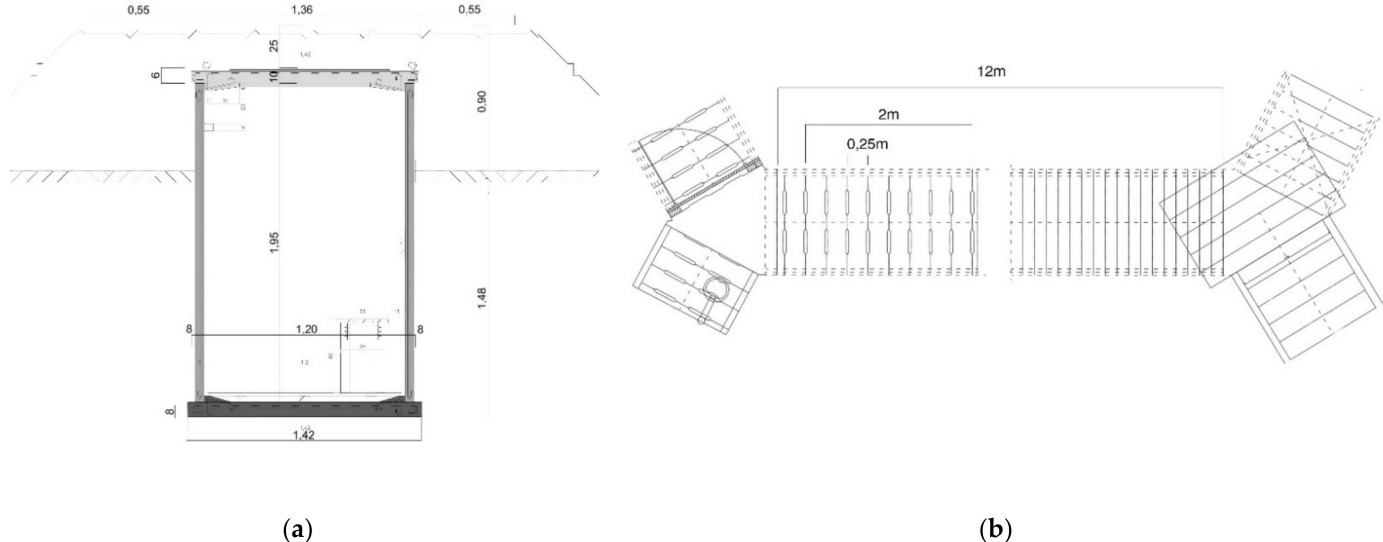

(**a**)                                                                                          (**b**)

**Figure 7.** Technical drawings based on archival documentation of prefabricated shelters. (**a**) Cross-section. (**b**) Modular plan.

All modules will have a common routing space with a raised floor to allow the use of the original drainage system located in the floor and a common space for a distributed ventilation system with inlets located in the lower part and an exhaust located in the upper part of the cabinet, ensuring continuity of the system along the entire length of the tunnel. The central section will house one of ten functions (Figure 8):

- Growing closet for large plants;
- Growing closed for medium plants;
- Growing closet for small plants;
- Growing closet for fungi;
- Aquaponic closet;
- Water and nutrition storage;
- General storage closet;
- Open shelves;
- Weighting, packing, and paying closet;
- Basket storage.

As with the design for the tunnels, the design of the entrance aimed to be as flexible and adaptive. The design consists of five elements that can be used depending on the type and degree of preservation of the structure. It takes the form of a modest addition that follows the shape of the original design. The new structure was deprived of its thickness and massiveness in order to create contrast and highlight the features of the existing structure. All the elements are placed inside, providing support without integration into the structure, to easily determine which parts should be used when an algorithm featuring a different scenario was created (Figure 9). It is worth noting that the system was created to secure the entrances in the cheapest and least aggressive way possible, and can be adapted or extended to allow more accessibility. Regardless of the situation however, each adaptation will add to the coherent presence of the adapted shelter system in the city tissue.

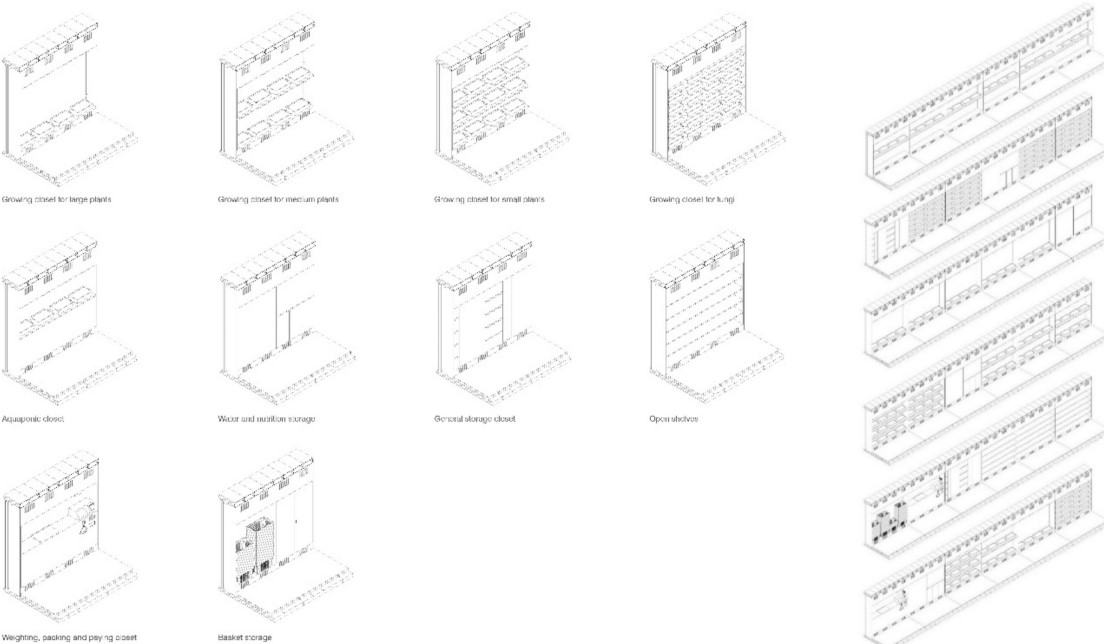

**Figure 8.** Specification of prefabricated modular equipment and an example of possible combinations of modules within one corridor.

### 3.6.3. The Human Factor—Activation of Neighbourhood Community

The idea of conversion into local farms is mainly based on social participation, and aims to strengthen the ties between members of the local community whilst at the same time creating a more prominent personal connection between citizens and their city by providing them with uniquely tailored spaces. This mutually beneficial symbiosis gives the opportunity to citizens to appropriate the leftover spaces of the public realm whilst creating a strong feeling of belonging at the same time, ultimately resulting in an increase in desire to care about their neighbourhood. The unique placement within the old neighbourhoods presents an opportunity to conduct a range of programs battling social exclusion. Large group of people inhabiting quarters of the city centre are seniors, struggling to be active in society. Many of them have skills to cultivate land and experience in vegetable growing thanks to the popularity of allotment gardening. Due to the physical character of work required for gardening and the shrinking number of allotment plots within the city, many had to give it up. Providing them with an opportunity to put their knowledge to use could help to rehabilitate them back into social life. Because of the proximity of schools, another socially excluded group prominent in the areas around the shelters is troubled teen. They are often seen as inexperienced and incompetent; furthermore, many are isolated from groups that are not their own peers, ultimately resulting in a struggle to be involved in social matters and decreasing their sense of belonging to the place they inhabit due to the lack of social connections outside of their age group. By giving opportunities to involve different age groups and stimulating their collaboration, the farm becomes a platform to battle exclusion and inequality caused by age stratification.

| | State of preservation | Used elements |
|---|---|---|
| 1. | | el.2, el.3, el.5 |
| 2. | | el.3, el.5 / el.2, el.3, el.5 |
| 3. | | el.3, el.5 / el.2, el.3, el.5 |
| 4. | | el.4 / el.4, el.1 |
| 5. | | el.4 / el.4, el.1 |
| 6. | | el.4, el.5 / el.4, el.1, el.5 |
| 7. | | el.4, el.5 / el.4, el.1, el.5 |

*el. 1 should be added when there is a need to shelter the area in front of the entrance or when the additional element is needed to cover the stairs.

*el. 4 should be added when the additional reinforcement is needed in the tunnel

The algorithm used to determine the usage of prefabricated elements for the construction of the entrance based on the current preservation state of the entrance pavilion

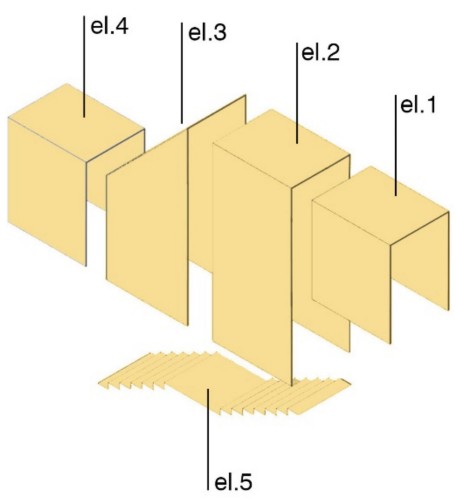

The simplified axonometric view of each of the lements used for the renovation of the shelter's entrances.

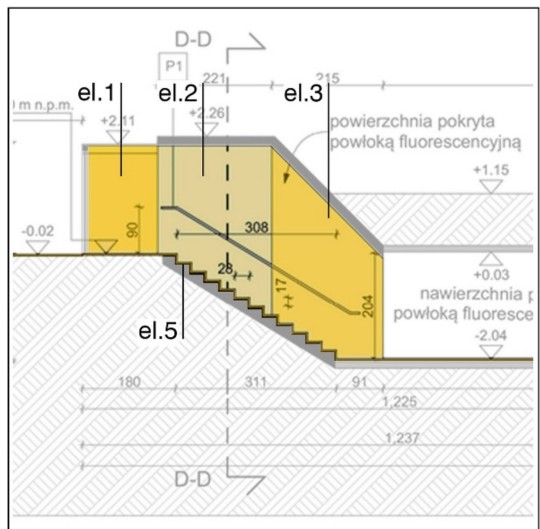

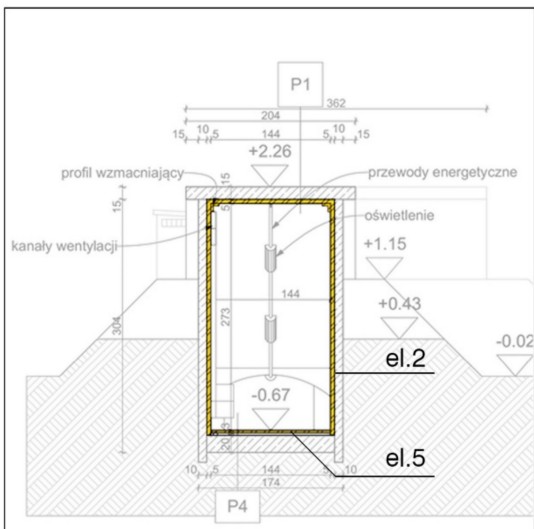

Fragments of the documentation for the project of adaptation of the shelter on Narutowicza street in Szczecin showing sections through the entrance pavillion. The prefabricated entrance elements - el.1, el.2 el.3 and el.5 used in the project were highlited and described.

**Figure 9.** Algorithm for designing the entrance and prefabricated elements description.

### 3.6.4. The Pilot Project and Feasibility Study

To confirm the flexibility of the system a pilot project was prepared. The location was chosen amongst existing shelters built within the city centre, and is placed in the green stripe dividing the roadways on Gabriela Narutowicza Street, between intersections with Głowackiego Street and Piastów Avenue. It is worthwhile to emphasise that neither the chosen location nor the facility itself is in any sense unique. Similar objects can be found

scattered around the entire city. Within a 500 m radius six similar shelters can be found; the closest is located on the same street, roughly 200 m away (Figure 10).

As with other locations, the shelter is mostly surrounded by residential buildings. In the quarter adjacent to the plot from the north, in addition to housing an international school is located, the area to the south belongs to a local university and houses the Faculty of Chemical Engineering and the Faculty of Maritime Technology and Transport, the area to the west consists mostly of small- and mid-sized enterprises, and the area to the east mainly consists of new residential buildings.

Each of the shelters is intended to be run by and serve its community. In this case, the location on the verge of old and new residential areas along with many schools and universities provides a wide range of potential users that possibly belong to different, disconnected social groups. It creates an opportunity for the shelter to function as a catalyst for the creation of new social bonds. The introduction of programs involving youth from nearby schools that could learn from the experience of the elderly that inhabit old dwellings could battle social exclusion of both groups. Programs aimed at the integration of young families from newly built housing developments with students could promote empathy and help to ease tensions. Building involvement could, with time, result in organic promotion of the farm and the creation of a sense of collective responsibility for its functioning.

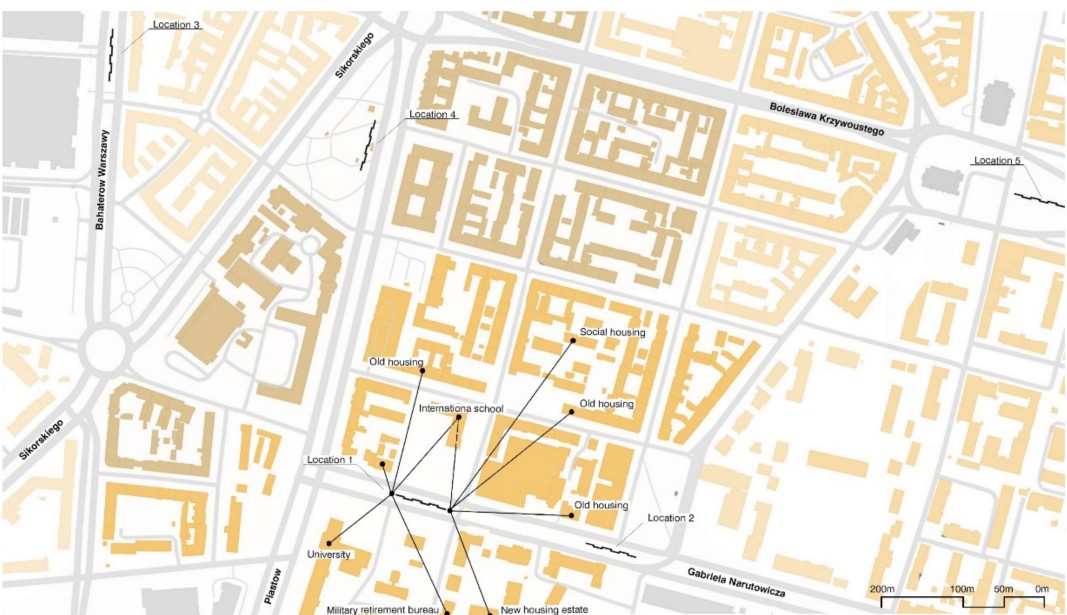

**Figure 10.** Shelters in the city centre and their neighbourhoods.

The proximity to the large supermarkets could possibly reduce the threshold for associated people, as they can firstly be supplied with the end product with the hope of future involvement. Providing sellable products is also crucial for the financial sustainability of transformation, where ultimately the profit from production covers the cost of operation and can possibly help in financing the revitalisation of another facility. It will allow the development of the system with a small initial outlay of funds. Via one-by-one development, a range of iterative design strategies can be implemented to test and adopt the system for newly discovered needs. Constant research and implementation will result in the evolution and improvement of its functioning.

Although there are no plans of this shelter, judging by the shape of the embankment, the system used for the construction of the entrance pavilion, and plans of other similar facilities in the city, it consists of four 12 m long corridors and two shorter corridors leading to the entrance pavilions. Each of the sections is built off the main axis and connected

by a short perpendicular segment. The entrance pavilions are left undamaged. Both lack original steel doors which were removed and replaced by a brick wall, sealing the entrance. The corridor itself most likely has a concrete barrel vault (Figure 11).

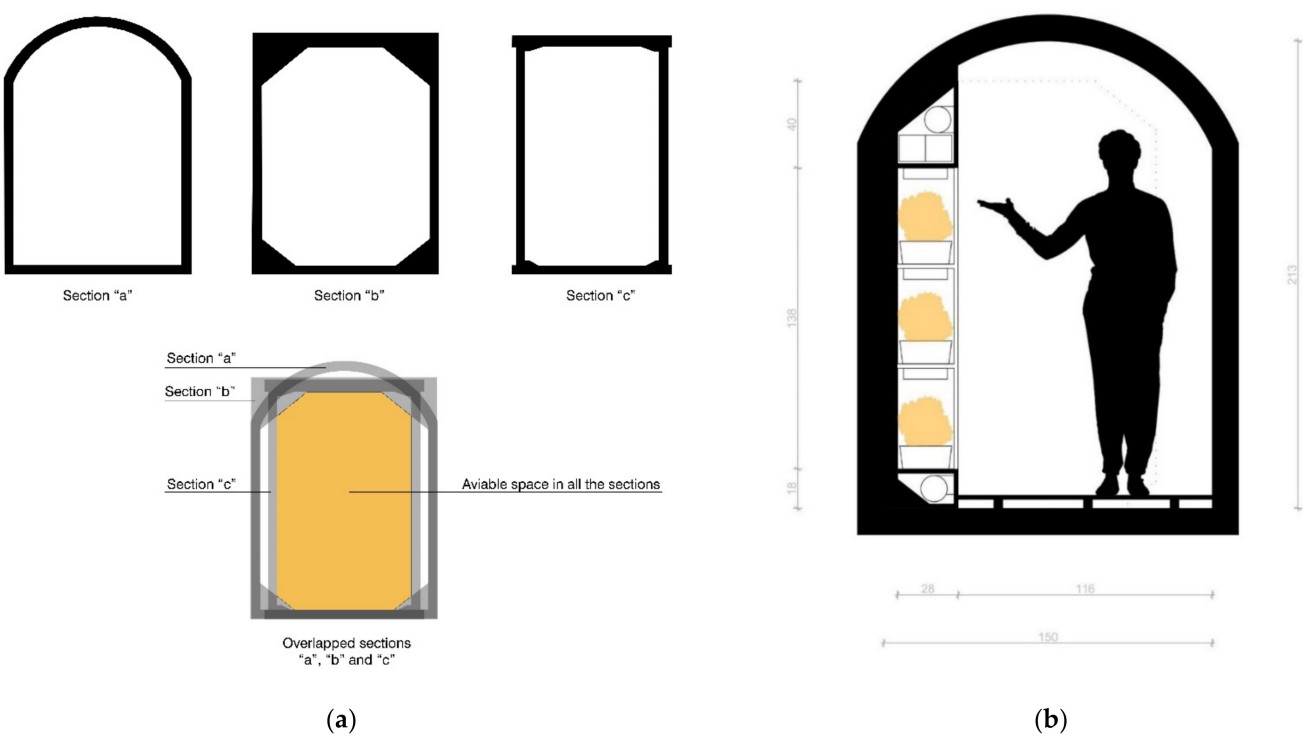

(**a**)

(**b**)

**Figure 11.** Sections: (**a**) overlapped section of different tunnels found in the city and (**b**) section of the designed tunnel.

The four main corridors will be equipped with 24 modules. Each of the corridors will be equipped with closets for the production of different sizes of crops. The eastern corridor will additionally host storages, shelves, tanks, and the module responsible for weighting, packing, and paying. Two short tunnels followed by the exits will remain empty, functioning as transitional space between the outside and the production area. The main door will be located between the first and second corridor, shifting the entrance threshold inside the structure; it reduces the pressure of involvement and gives room for the development of social interactions. The entrance pavilions are kept in perfect condition, and they allow for the use of the first option from a designed algorithm, using two elements that will secure the stair area. Throughout the whole length of the shelter a single material path will be laid that will extend far beyond the shelter until it intersects with a pedestrian path to strengthen its presence in the urban environment, which in the future can serve as a strong brand-building base (Figure 12).

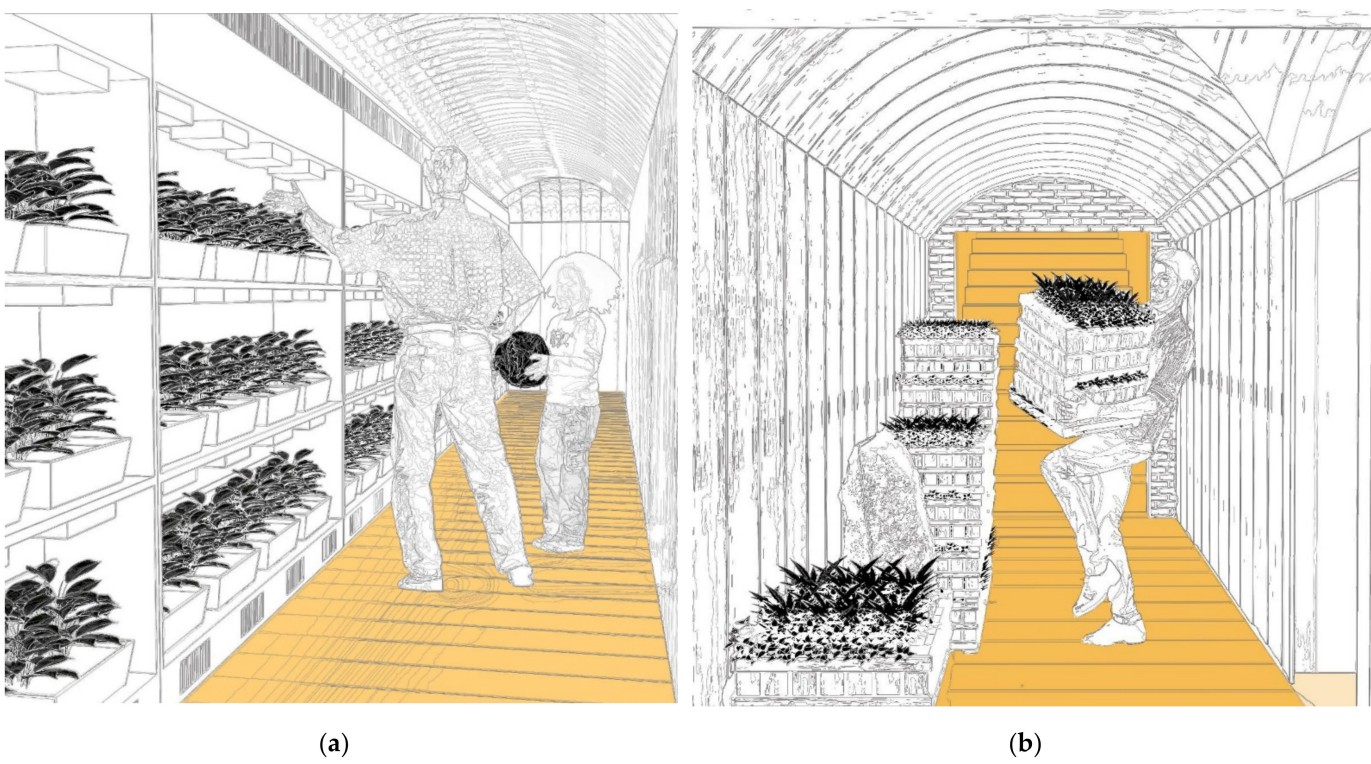

(**a**)                                                                          (**b**)

**Figure 12.** Visualisations. (**a**) Production area. (**b**) Entrance tunnel.

## 4. Discussion

The benefits of the construction of underground hydroponic urban farms include many positive environmental aspects, as the production requires no soil and does not utilise ground surface space, which in case of large-scale production can impact local ecosystems. The production also uses considerably less water, which globally is a diminishing resource, and results in much higher yields compared to traditional systems. It is important to point out that although these are strong arguments, the production in shelters will never be as efficient as newly designed factories, and the scale of proposed production will have nearly zero impact on the battle with global shortages of soil and water. It will, however, add to environmental education, which can affect future efforts in battling the aforementioned global problems. Introduction of the system will also bring a stable and independent source of vegetables on a neighbourhood scale. Location of the production underground will allow for year-long crops with minimum energy consumption for heating or cooling purposes, as the temperature inside of the tunnels is stable throughout the year. The highly controlled environment can also protect the crops from natural disasters such as floods, droughts, pests, or disease attacks. To maintain that level of protection the hydroponic system requires maintenance and constant control, as the isolated environment, lacking a natural soil buffer, is much more vulnerable to infestations, system malfunctions, lack of nutrients, or power outages. Operation of the system by the local community can be both advantage and disadvantage in that matter. The production will be most likely monitored by many people with no experience with such productions before. Planning and strategic involvement of experienced personnel will be crucial in the early stages of the revitalisation.

From an economic standpoint the proposed underground farms also have strong pros and cons. The initial costs of setting up the production will much likely be lower than the setup of similar production in the newly built space, thanks to reuse of existing underground space, but will be much higher compared to traditional farms. It will, on the other hand allow, more control over underground corridors and prevent unexpected collapses and damages to the infrastructure of cities. In the past there were multiple examples of roads and pavements or even building damages caused by the collapse of

poorly maintained underground shelters. The high costs of production on such a small scale could be brought down by providing sustainable sources of electricity, considering a much shorter supply chain and nearly cost-free space and volunteer work of the local community. The overall cost of produced vegetables will also decrease over time as involved representatives of local communities will gain knowledge in operating the system.

For visitors and users, the indoor environment of an underground shelter is an ideal option for every season of the year. In spring and summer, they offer a moment's respite from the scorching heat; in autumn and winter, they are a perfect time-filler and diversification. No matter what the weather is—rain or frost—there is usually a constant temperature. Underground hydroponic cultivation can act as a kind of isolated sanctuary, as 99% of the underground is out of mobile phone range, which can be an important feature of space use today [46] Recognising that every innovative social project, new initiative, or emerging novel technology raises doubts, numerous questions, and controversies, an open list of pros and cons of the proposed regeneration project is left for discussion, further deliberation, and detailed research (Table 2).

**Table 2.** Comparison of pros and cons for project implementation.

|  | Advantages | Disadvantages |
|---|---|---|
| 1 | Independent food production | Energy consuming |
| 2 | High yields with minimum space | Vulnerable to power outages |
| 3 | Controlled environment | If infected, diseases spread faster |
| 4 | No use of pesticides and harmful chemicals | Community base may be unreliable |
| 5 | Minimal chance of pest infections and diseases | Small yields compared to maintenance time compared to big farms |
| 6 | Minimum building costs thanks to reuse of existing spaces | High costs of production |
| 7 | Low heating costs | High start-up costs |
| 8 | Constant production regardless of season and climate |  |
| 9 | Education |  |
| 10 | Possible community benefits |  |
| 11 | Short supply chain |  |
| 12 | Saves resources |  |

History has been invoked and presented as heritage—as a significant past that should be remembered; more and more buildings and other places have been called upon to act as witnesses to the past. Many types of groups have sought to secure public recognition by identifying and presenting "their" heritage.

By looking at heritage that is disturbing and inconvenient, rather than that which can be celebrated or at least recognised as part of the cherished history of a nation or city, it is important to avoid cynical selectivity and focus on the ecological effectiveness of the material resources at hand. The potential of existing underground structures, such as the presented system of underground air raid shelters in Szczecin, should be exploited, and their accessibility to local communities should be taken into account. A replicable hydroponic food production system, linked to modular urban aquaponics, will perfectly fill the space of the prefabricated tunnelled underground shelters. The central location of the shelters in relation to the current housing stock encourages the development of such urban farms, establishing local urban farming communities and activating neighbourhood communities. Such initiatives foster an increase in the level of food sovereignty, improve the social security of neighbouring residents, and increase the biological productivity of urbanised areas in the spirit of regenerative economics. It must be affirmed that from a technical and technological point of view the proposed systematic regeneration of bunkers in the current community structure of contemporary Szczecin is possible. Such an initiated regeneration creates potential for building new relations between citizens and the complex history of their city.

## 5. Conclusions

Taken together, the above data may lead to the conclusions that, for the success of future redevelopment, planners should focus on providing functions that could be used

according to the original neighbourhood range of the entire infrastructure, rather than basing the choice of a new function on a narrow, exclusive group of users with specific interests or wealth. Restoring the communal character of shelter use in a new, inclusive way should be one of the main objectives of future regeneration attempts.

The concept of decentralized urban underground farming as a part of urban agriculture has both strong and weak points. We should remember, however, that the main reason for the revitalisation lies in the historical quality of the space itself. The benefits of preserving historical structures of such importance, which until this day sparks strong emotions in the inhabitants of the city, may outweigh any inconveniences caused by situating farming, or any other function in the shelter system. The advantages of a social-oriented initiative may also add to the arguments in favour for the revitalisation. If successful, the benefits of an integrated society with many interlinking social connections within itself that developed a strong bond with the place they are inhabiting will be in long term more valuable for the city than any costs caused by the development of the project.

**Author Contributions:** Conceptualisation, P.M. and L.Ś.; methodology, P.M. and L.Ś.; software, P.M.; validation, L.Ś.; formal analysis, P.M.; investigation, P.M.; resources, P.M. and L.Ś.; data curation, P.M.; writing—original draft preparation, P.M.; writing—review and editing, L.Ś.; visualisation, P.M.; supervision, L.Ś.; project administration, L.Ś.; funding acquisition, L.Ś. All authors have read and agreed to the published version of the manuscript.

**Funding:** This research received no external funding.

**Institutional Review Board Statement:** Not applicable.

**Informed Consent Statement:** Not applicable.

**Data Availability Statement:** Data are contained within the article or available from referenced sources.

**Conflicts of Interest:** The authors declare no conflict of interest.

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
