# Peer review of "The Unwanted Heritage of Prefabricated Wartime Air Raid Shelters—Underground Space Regeneration Feasibility for Urban Agriculture to Enhance Neighbourhood Community Engagement"

_sustainability, doi:10.3390/su132112238_

Round 1

Reviewer 1 Report

Dear authors,

Congratulations on choosing this particular topic; I found it very interesting and valuable due to its representativity for the academic community devoted to studying heritage and urban planning. 

However, the text has to go through a major revision in order to fit the requirements of the journal you chose for publication.

Therefore, I would have several suggestions:

  1. Adjust the title - since the most significant part of the study focuses on using the prefabricated spaces of the historical air-raid shelters as “underground gardens” for civic activation, it should be clear from the very beginning.
  2. The introduction needs to be improved with more references. Mark the most important events on the graph presenting the evolution of population for a better understanding (figure 1b). Mark the underground civil defence infrastructure evolution in a figure or table for better understanding.
  3. Improve the “materials and methods” section. Please present a clear sequence of the steps you followed in generating the data used for interpretation. Insert information regarding the software used for generating the sketches. Please rewrite the phrase rows 117-119, the topic needs to be changed.
  4. Reorganise the “results” section for better readability. Use headers and subheaders and please provide more references since there are many paragraphs lacking them (eg. row 279-289). Present the examples of good practices in a highlighted way (tables or figures) to improve readability. 
  5. The “discussion” section starts very abruptly, please create a coherent link between the topics.
  6. The paper has no conclusions.

Hoping you will proceed with the changes abovementioned, I wish you good luck with the studies you are conducting.

Author Response

We are grateful for the reviews received, the valuable suggestions and the reviewers' suggestions. We have tried to improve the initial structure of the text in order to make the message of the article and the method of argumentation used in our project more clear.

Below we refer to the detailed comments listed in the individual review reports, hoping that we have at least partially met the expectations of the editors and esteemed reviewers to accept our article for publication.

Review Report 1

Date of this review 18 Aug 2021

Ad. 1:

“Adjust the title - since the most significant part of the study focuses on using the prefabricated spaces of the historical air-raid shelters as “underground gardens” for civic activation, it should be clear from the very beginning.”

Answer:

The suggestion to change the title was very inspiring and accurate, which prompted us to reformulate the title and make it read as follows: “The Unwanted Heritage of Prefabricated Wartime Air Raid Shelters - an Underground Spaces Regeneration Feasibility for the Urban Agriculture to Enhance Neighbourhood Community Engagement”.

Ad. 2:

The introduction needs to be improved with more references. Mark the most important events on the graph presenting the evolution of population for a better understanding (figure 1b).

Mark the underground civil defence infrastructure evolution in a figure or table for better understanding.”

Answer:

We have introduced additional reference positions both in the introduction and in the whole text.

The key events like the beginning and the end of the II World War were added to the graph (Figure 1b). The important periods like Industrialization which led to rapid increase in population of the city and to increase in the strategic importance of the city, and period of post-war repopulation of Szczecin.

We have added an additional figure (new Figure 4) showing the development of the defence infrastructure system in Szczecin. The figure shows graduate expansion and decentralization of the defence infrastructure system.

Ad. 3:

Improve the “materials and methods” section. Please present a clear sequence of the steps you followed in generating the data used for interpretation. Insert information regarding the software used for generating the sketches. Please rewrite the phrase rows 117-119,

Answer:

The “Materials and Methods” section was improved with description about data generation and interpretation.  We have included the information about the used software in the final part of this section.

The sentence (the phrase rows 117-119) has been rewritten. We want to highlight that accessibility of information can be a determining factor in any attempts of revitalization as not all the Architecture offices or bottom up initiatives have budgets for extensive archive research. We wish for this publication to be a bridge and provide information about the underground system to whoever will try to adapt those spaces in the future.

Ad. 4:

“Reorganise the “results” section for better readability. Use headers and subheaders and please provide more references since there are many paragraphs lacking them (eg. row 279-289). Present the examples of good practices in a highlighted way (tables or figures) to improve readability”.

Answer:

We have added additional headers to increase the readability. We have also redraw and included the original plans from the archive illustrating the most complex part which is the spatial description of the shelter and the prefabricated used for its construction and included pictures of the current state of the shelters. We have also simplified the algorithm presented in Figure 6. (revised Figure 9.) and extended and repositioned the paragraph describing the current state of the shelters.

More reference items have been added.

The structure of the text of the section on good practice has changed. We have introduced sub-headings which we hope will clearly organise the reading of the good practice examples.

Ad. 5:

“The “discussion” section starts very abruptly, please create a coherent link between the topics.”

Answer:

The chapter called “Discussion” in the initial version has been restructured. A substantial part of this chapter now has a new subtitle: “3.6. Proposal description of prefabricated wartime air shelters regeneration.” The changes made are expected to ensure that the different strands of the text are coherently linked in order to improve the readability of the arguments.

Ad. 6:

“The paper has no conclusions.”

Answer:

We have suggested a short chapter containing the conclusion of our article.

We hope that after taking into account the reviewers' valuable comments and after making appropriate corrections to the text, our article has become more readable and convincing.

We believe that the proposals and results presented in the article and the attempt to generate scientific discussion will be interesting for the readers of this journal.

Reviewer 2 Report

The article presents a proposal for the sustainable reuse of underground shelters in Szczecin, Poland. The article is well-structured, and the topic approached is current and relevant. The proposed idea also looks interesting, although it lacks more detail.

However, section 4 should be renamed as it is not the discussion of the article. The content of this section is the proposal description, which needs to be further justified (“urban agriculture fits best” is clearly insufficient). Authors should write a new section that discusses the advantages and disadvantages of the proposed option compared to other space reuse alternatives. Figure 6, besides being incomprehensible, has no reference in the text describing the algorithm. What does it consist? What is it for? What are the results?, etc.

Other details to improve:

  • Please standardize references (e.g. in lines 35-36 a different referencing procedure is used);
  • Please order the references as they are called in the text (e.g. the references [6], [7], and [8] are called before the reference [4]);
  • Please include references in some statements (e.g. lines 92-96);
  • Please include figures illustrating the descriptions (e.g. lines 439-449);
  • Please include captions in the text to figures 5 to 9;
  • Figure 7: The location description (lines 510-516) without the street locations in the figure is irrelevant.
  • Figure 8 (a): What does this figure mean?

Author Response

We are grateful for the reviews received, the valuable suggestions and the reviewers' suggestions. We have tried to improve the initial structure of the text in order to make the message of the article and the method of argumentation used in our project more clear.

Below we refer to the detailed comments listed in the individual review reports, hoping that we have at least partially met the expectations of the editors and esteemed reviewers to accept our article for publication.

Review Report 3

Date of this review 22 Sep 2021

Ad. 1:

“However, section 4 should be renamed as it is not the discussion of the article. The content of this section is the proposal description, which needs to be further justified (“urban agriculture fits best” is clearly insufficient). Authors should write a new section that discusses the advantages and disadvantages of the proposed option compared to other space reuse alternatives.”

Answer:

In the completed text we have presented a list of possible adaptations and functional solutions for the analysed network of prefabricated air-raid shelters, with an indication of systemic solutions that can meet the needs of neighbouring communities. Considering only complex and systemic solutions, fitting the specific geometrical configuration and dimensions of a large number of prefabricated shelters scattered in the urban tissue, we came to the conviction that the adaptation of underground spaces for the function of urban hydroponic cultivation is the most adequate for a positive regeneration process.

Ad. 2:

“Figure 6, besides being incomprehensible, has no reference in the text describing the algorithm. What does it consist? What is it for? What are the results?, etc.’

Answer:

Figure 6 (revised Figure 9.) showing the algorithm has been redesigned and annotated. The illustration was divided into separate graphics. In the first one, the designed elements of the entrance were given numbers from el.1 to el. 5. The next graphic shows the arrangement of the elements on the part of the technical documentation of the pilot project of the shelter adaptation.

Ad. 3:

Please standardize references (e.g. in lines 35-36 a different referencing procedure is used);

Please order the references as they are called in the text (e.g. the references [6], [7], and [8] are called before the reference [4]);

Please include references in some statements (e.g. lines 92-96);

Answer:

We apologise for our oversight. This particular type of reference has been corrected and standardised in accordance with the reference procedure. A new order has been introduced for all reference items. Several new reference items have been introduced in specific parts of the text

Ad. 4:

Please include figures illustrating the descriptions (e.g. lines 439-449);”

Answer:

We have added drawings extracted from original documentation of the shelter with the smallest section, one determining the shape of the modules the most in section 3.6. of the text. (new Figure 7. ). 

We have also added an additional illustration (new Figure 8.) presenting each of the modules complementing the list in rows 459 - 468.

Ad. 5:

“Please include captions in the text to figures 5 to 9;”

Answer:

We have included captions to all the figures used in the article. New figures also have their captions in the text.

Ad. 6:

“”Figure 7: The location description (lines 510-516) without the street locations in the figure is irrelevant.”

Answer:

The street names have been added to the figure (revised Figure 10.). We have also added the names of the key facilities in the nearby surroundings that were used to plan the functioning of the urban farm and marked other locations of the nearby shelters. Additionally we also included the scale.

Ad. 7:

“Figure 8 (a): What does this figure mean?”

Answer:

The figure has been redrawn. It now clearly shows the 3 sections that were overlapped to determine the common space available in the corridors. This figure (revised Figure 11) was crucial for the design process of the modules as it showed us how much maximum space can one module occupy and still fit in all the shelters.

Thank you for your rightful comment. The original version of the graphic was too poor to clearly communicate the result of our analysis.

We hope that after taking into account the reviewers' valuable comments and after making appropriate corrections to the text, our article has become more readable and convincing.

We believe that the proposals and results presented in the article and the attempt to generate scientific discussion will be interesting for the readers of this journal.

Reviewer 3 Report

The paper is an original one, since it deals with the contradictory and controversial legacy of air raid shelters in the Polish city of Szczecin, a German city at the time of the Second World War.
'Upcycling' such huge underground heritage, as the authors argue, overcoming painful pages of past history, can be tied to hydroponic agriculture that are inscribed in the memory or in the wishes of the inhabitants.

The quality and scientific soundness of the work is  good, yet some explanations are required for improving the paper:

the position of Municipality towards this hypothesis is not mentioned. Who would be in charge of the of the project, subsequent implementation, and of running the activities?

Is it possible to have some pictures of the current state of entrances in the urban fabric?

Author Response

The article presented here focuses on the heritage associated with the Nazi German underground air raid shelter system and the eco-efficient revitalisation objectives to encourage local communities to accept and make creative use of the existing structures. As a result of the analysis and research carried out, systemic, modular and replicable solutions have been proposed to enable local hydroponic and aquaponics cultivation as part of Urban Agriculture. The sun was symbolically introduced into underground spaces burdened by the lack of a view of the blue sky. This allows building food security and sovereignty of the local community, functioning in the immediate vicinity of air-raid shelters - silent witnesses of the history of the Second World War.

We are grateful for the reviews received, the valuable suggestions and the reviewers' suggestions. We have tried to improve the initial structure of the text in order to make the message of the article and the method of argumentation used in our project more clear.

Below we refer to the detailed comments listed in the individual review reports, hoping that we have at least partially met the expectations of the editors and esteemed reviewers to accept our article for publication.

Review Report 2

Date of this review 16 Sep 2021

Ad. 1

“The position of Municipality towards this hypothesis is not mentioned. Who would be in charge of the of the project, subsequent implementation, and of running the activities?”

Answer:

The position of the municipality is indeed not mentioned, as there is no official statement regarding this research. We have spoken with municipality representatives in private and got their opinion on the matter. Unofficially we have been told that the city favours such initiatives (at least on a small scale) and we were explained the legal process regarding the occupation of the underground spaces. 

As issuing any official statement on municipal level regarding large scale proposals would have political consequences, hence would be complicated and time consuming to obtain, we decided to focus our efforts on other areas of this research. However, I absolutely agree that answering those questions is extremely important, and I would be thrilled to be able to open a dialog with the municipality about it.

When it comes to running the activities we put much hope in self-governance of local communities. We believe that coming up with the exact scheme for operation of such a place will differ depending on the place of adaptation. We would love to see education facilities or senior clubs getting involved and appropriating the underground spaces. In each case we would need an organization that could be a legal entity to control the revitalization but we do not specify whether that should be one organization partially controlled by the government or whether it should be a decentralized net of privately owned NGO’s or perhaps even corporations.

Ad. 2

“Is it possible to have some pictures of the current state of entrances in the urban fabric?”

Answer:

We have included the pictures (new Figure 3) of the well preserved entrance of the shelter in “location 3” (revised Figure 10) in the urban fabric and the picture from the inside of the pavilion. In the text we have added some additional information about the current state of the bunkers in the context of the surroundings.

We hope that after taking into account the reviewers' valuable comments and after making appropriate corrections to the text, our article has become more readable and convincing.

We believe that the proposals and results presented in the article and the attempt to generate scientific discussion will be interesting for the readers of this journal.

Round 2

Reviewer 1 Report

Dear authors,

I consider the changes you made have significantly improved the quality of the paper. 

However, it is still quite long and highly descriptive. I would suggest transforming the detailed 9 steps from chapter 2. Materials and Methods into a table and restructuring a bit the methodology.

Please remove the reference from the conclusions, there should be no quotations there.

Good luck with your further research!

Author Response

Thank you for your interesting comments and suggestions. In Chapter 2 Materials and Methods we have introduced a table: "Selected research methods and objectives". Some of the too detailed information and comments have been eliminated. We hope that this will improve the proportion of the whole article and make it easier to read. In the conclusion we have, of course, dispensed with quotations. We are aware of a certain imperfection of the presented article, however, we hope that the described research problem and suggestions for solving it are clear and will be interesting for a wide range of readers.